# How Essential Kinesin-5 Becomes Non-Essential in Fission Yeast: Force Balance and Microtubule Dynamics Matter

**DOI:** 10.3390/cells9051154

**Published:** 2020-05-07

**Authors:** Masashi Yukawa, Yasuhiro Teratani, Takashi Toda

**Affiliations:** 1Laboratory of Molecular and Chemical Cell Biology, Graduate School of Integrated Sciences for Life, Hiroshima University, Higashi-Hiroshima 739-8530, Japan; m185804@hiroshima-u.ac.jp; 2Hiroshima Research Center for Healthy Aging (HiHA), Hiroshima University, Higashi-Hiroshima 739-8530, Japan

**Keywords:** bipolar mitotic spindle, fission yeast, kinesin, kinetochore, microtubule dynamics, microtubule polymerase, microtubule–associated proteins (MAPs), spindle pole body (SPB), sister chromatid cohesion

## Abstract

The bipolar mitotic spindle drives accurate chromosome segregation by capturing the kinetochore and pulling each set of sister chromatids to the opposite poles. In this review, we describe recent findings on the multiple pathways leading to bipolar spindle formation in fission yeast and discuss these results from a broader perspective. The roles of three mitotic kinesins (Kinesin-5, Kinesin-6 and Kinesin-14) in spindle assembly are depicted, and how a group of microtubule-associated proteins, sister chromatid cohesion and the kinetochore collaborate with these motors is shown. We have paid special attention to the molecular pathways that render otherwise essential Kinesin-5 to become non-essential: how cells build bipolar mitotic spindles without the need for Kinesin-5 and where the alternate forces come from are considered. We highlight the force balance for bipolar spindle assembly and explain how outward and inward forces are generated by various ways, in which the proper fine-tuning of microtubule dynamics plays a crucial role. Overall, these new pathways have illuminated the remarkable plasticity and adaptability of spindle mechanics. Kinesin molecules are regarded as prospective targets for cancer chemotherapy and many specific inhibitors have been developed. However, several hurdles have arisen against their clinical implementation. This review provides insight into possible strategies to overcome these challenges.

## 1. Introduction

### 1.1. Bipolar Mitotic Spindles and Kinesin Motor Proteins

The bipolar mitotic spindle is a dynamic ensemble of core microtubule polymers and a cohort of microtubule-associated proteins (MAPs). It attaches to the kinetochore on sister chromatids to align chromosomes at the spindle equator in metaphase, and pulls each pair of sister chromatids towards the opposite poles in anaphase A. During anaphase B, spindles further elongate to ensure the equal partition of each set of chromosomes, which is followed by cytokinesis [1,2]. MAPs are required for spindle assembly and coordinate the multiple events of mitosis in a spatiotemporal manner. Kinesin motors comprise one of the major families of MAPs and couple the energy of ATP hydrolysis to force generation [3]. The kinesin superfamily comprises at least 15 families, which are further structurally categorised into three groups, called N-kinesin (N-terminal), M-kinesin (middle) and C-kinesin (C-terminal), depending on the location of the motor domain within each molecule [4,5,6]. Mitotic kinesins include 10 kinesin families that are functionally designated as they are localised to the spindle microtubule and regulate structure and function of the spindle and mitotic progression [7]. This review is based upon recent work using fission yeast as a model but includes comparisons with other systems, in which evolutionary conservation and diversification are discussed.

### 1.2. Kinesin-5 Plays an Essential Role in Bipolar Spindle Assembly and Cell Survival

The type 5 kinesin (Kinesin-5) was originally identified in *Aspergillus nidulans* as one of the mitotically arrested mutants (called *bimC*) [8]. This kinesin belongs to the N-kinesin that moves on microtubules towards their plus ends. This motor forms homotetramers, thereby crosslinking and sliding apart antiparallel microtubules [9,10]. During early mitosis, this process generates an outward pushing force towards two duplicated spindle poles (centrosomes in animal cells and the spindle pole bodies (SPBs) in fungi), which promotes centrosome/SPB separation, thereby establishing spindle bipolarity [11,12]. In most, if not all, eukaryotes, Kinesin-5 (budding yeast Cin8 and Kip1, fission yeast Cut7, *Aspergillus* BimC, *C. elegans* BMK-1, *Drosophila* Klp61F, *Xenopus* Eg5 and human KIF11) is essential for mitosis, in which any means of its inactivation, e.g., chemical inhibition, genetic deletion or RNAi-mediated depletion, leads to the emergence of monopolar spindles, the failure of chromosome segregation and viability loss [8,13,14,15,16,17,18,19].

## 2. How Essential Kinesin-5 Becomes Non-Essential

Surprising findings came to light that cells can divide in the absence of Kinesin-5 function under certain conditions across a wide range of species, including human beings, frogs, flies, filamentous fungi and the budding and fission yeasts. Initial genetic studies and recent more comprehensive analysis conducted in *Aspergillus nidulans* and *Saccharomyces cerevisiae* showed that lethal mutations in Kinesin-5 are rescued by simultaneous inactivation of genes encoding Kinesin-14 (*klpA* and *KAR3* respectively) [20,21,22,23]. This is because bipolar spindle assembly is driven by the finely tuned, antagonistic force balance exerted by opposing motors. More precisely, an outward force generated by plus end-directed Kinesin-5 is antagonised by an inward force produced by minus end-directed Kinesin-14 that belongs to the C-kinesin (budding yeast Kar3, fission yeast Pkl1 and Klp2, *Aspergillus* KlpA, *Drosophila* Ncd, *Xenopus* XCTK2 and human HSET/KIFC1) or Dynein [24]. Accordingly, inactivation of minus end-directed motors could neutralise the loss of Kinesin-5 activity. In other words, cells without Kinesin-5 and Kinesin-14 or Dynein are now capable of forming bipolar spindles and will continue to divide.

## 3. Conditions Under Which Cells Do Not Need Kinesin-5 for Survival

As aforementioned, the main means in which Kinesin-5 becomes dispensable is by simultaneous inactivation of opposing Kinesin-14 or Dynein. In order to explore the genetic network that plays a role in conferring the non-essentiality of Kinesin-5, we conducted systematic screening for suppressors against *cut7* temperature sensitive (ts) mutants in fission yeast. Spontaneous survivors of *cut7* mutant strains grown at the restrictive temperature (36 °C) were isolated. After standard genetic analyses and nucleotide sequencing, suppressor genes (designated *skf* = *s*uppressor of *k*inesin *f*ive) were identified (Table 1) [25]. Suppressors can be classified into three main groups: Kinesin-14s, non-motor MAPs and tubulins.

### 3.1. Suppression by Mutations in Kinesin-14s or Their Cofactors 

In fission yeast, two Kinesin-14s, Pkl1 and Klp2, form distinct complexes with specific cofactors and are localised to different sites on the microtubule to play non-redundant roles in spindle assembly and mitotic progression [26,27,28,29]. Pkl1 forms a ternary complex with Msd1 and Wdr8 (referred to as the MWP complex) and is localised predominantly to the mitotic SPB, thereby anchoring the minus end of the spindle microtubule to the SPB [30,31,32]. During early mitosis, when the duplicated SPBs start to separate in a process driven by the Cut7-mediated outward force, the SPB-tethered MWP complex is loaded onto the spindle microtubule that nucleates from the other SPB, where this complex exerts an antagonistic inward force. Thus, the reason for suppression of *cut7* mutants by *pkl1∆*, *msd1∆* or *wdr8∆* is that the two SPBs can separate because the overwhelming inward force exerted by the MWP Kinesin-14 complex disappears (Figure 1A).

By contrast, Klp2 is mainly localised along spindles in a punctate manner [34]. Spindle-localising Klp2 crosslinks the antiparallel microtubule bundles, which produces an inward force by exploiting minus-end motility, and this force acts antagonistically with the Cut7-driven outward force on the spindle microtubule [33]. The reason for suppression of *cut7* mutants by *klp2∆* is that antiparallel microtubules can elongate in the absence of Cut7, as Klp2-driven inhibitory inward force is lost (Figure 1B). Collectively, Pkl1 acts mainly during the early stages of bipolar spindle assembly, while Klp2 plays a role in spindle elongation at later stages of mitosis. These distinct modes of the spatiotemporal regulation between Pkl1 and Klp2 underlie the collaborative actions of these two Kinesin-14s.

The deletion of either *pkl1* or *klp2* suppresses the temperature sensitivity caused by the *cut7* mutations [27,29,35]. Intriguingly, gene deletion of *pkl1*, but not *klp2*, is capable of rescuing a complete deletion of *cut7* [33,36,37,38]. Despite apparent ordinary growth, *cut7∆pkl1∆* cells display mitotic delay in which cells spend a longer period of time with short spindles. This implies that in the absence of Cut7 and Pkl1, an excessive inward force is imposed during early mitosis. This inward force, at least in part, is generated by Klp2, as the slower spindle elongation rate is significantly ameliorated in the *cut7∆pkl1∆klp2∆* triple mutant [33].

### 3.2. Suppression by Compromised Microtubule Nucleation, Polymerisation and Stability

We have found that mutations in the genes encoding tubulins and five non-motor MAPs are also capable of rescuing *cut7* ts mutants. In fission yeast, tubulin molecules are encoded by *nda2* (α1-tubulin), *atb2* (α2-tubulin) and *nda3* (β-tubulin) [41,42] (Table 1). Mutations in tubulin genes would compromise microtubule integrity. One of the five MAPs is Mal3/EB1, a conserved MAP that tracks on the microtubule plus end [43]. Its mutation leads to microtubule destabilisation and defects in kinetochore–microtubule attachment [44,45]. Alp16 is a homologue of GCP6 and a component of the microtubule-nucleator γ-tubulin complex (γ-TuC) [46,47,48]. The other three MAP-encoding suppressors are *alp7* (encoding an orthologue of the transforming acidic coiled-coil protein TACC) [49,50], *alp14* and *dis1* (two genes encoding XMAP215/Stu2/TOG microtubule polymerases) [51,52,53,54,55,56]. Alp7 and Alp14 form a stable complex in the cell and promote microtubule polymerisation [49,54,57]. The Alp7–Alp14 complex is also required for efficient nucleation of the microtubule from the SPB through interaction with the γ-TuC [58]. 

Apart from its role in microtubule stabilisation, Mal3/EB1 is known to interact with Klp2, which is a prerequisite for this motor to be loaded on the spindle microtubule [34]. Thus, suppression of the *cut7* ts mutation by the *mal3* mutation could be ascribable to the loss of Klp2 function [25]. Overall, the common features of suppressor genes encoding tubulins and MAPs are that all these mutations lead to the destabilisation of the spindle microtubules. It is worth noting that in *cut7* mutant cells, the intensities of spindle microtubules are augmented [25]. Importantly, in these mutant cells, intensities of Klp2 on the spindle microtubule are also substantially increased. Notably, in the double mutant containing *cut7* ts and any of the suppressor mutations, the intensities of Klp2 are reduced. Given these observations, we propose that the rescue of *cut7* mutants by these suppressors is derived from quantitative downregulation of Klp2 activity. In fact, the overproduction of Klp2 in the double mutants between *cut7* and suppressor mutations restored a ts phenotype similar to a single *cut7* mutant [25], corroborating the notion that the reduced localisation/activity of Klp2 is the primary, if not the sole, reason for the rescue of the *cut7* mutation. In summary, suppressor analyses have uncovered that multiple factors that regulate microtubule structures are involved in several mechanisms by which Kinesin-5/Cut7 becomes dispensable, and importantly, inactivation of Kinesin-14s, Pkl1 and Klp2, is the main means for the rescue of *cut7* by these suppressors. 

### 3.3. Suppression by Microtubule-Destabilising Drugs

As previously mentioned, *cut7* mutants exhibit increased intensities of the spindle microtubule accompanied by more Klp2 proteins on the spindle microtubule. In line with this observation, these cells display hyper-resistance against microtubule-depolymerising drugs, such as thiabendazole (TBZ) or methyl 2-benzimidazolecarbamate (MBC), and interestingly, treatment of *cut7* ts mutants with TBZ or MBC rescues temperature sensitivity [25]. Under this condition, Klp2 levels are lessened as in the other suppressor mutations. Remarkably, drug treatment rescues even a complete deletion of *cut7*. Collectively, the impairment of microtubule stability and/or dynamics by either suppressor mutations or treatment with microtubule-destabilising drugs renders fission yeast cells viable in the absence of Kinesin-5 (Figure 2). 

It is worth pointing out that in cultured human cells (HeLa or U2OS), microtubule destabilisation can also effectively rescue monopolar spindle phenotypes induced by Kinesin-5/KIF11 inhibition [59,60]. This underscores the evolutionary conservation of Kinesin-5 function and its regulation from fission yeast to human beings.

### 3.4. Suppression through Downregulation of the cAMP/PKA Pathway

In many organisms, the extracellular environment, such as nutritional cues, regulates microtubule dynamics through intracellular signal transduction pathways. In yeasts, glucose in the media activates the cAMP/PKA pathway, by which it controls a diverse set of downstream events [61,62,63]. Deletion of the *pka1* gene rescues the *cut7* ts mutant [64]. Pka1 reportedly fine-tunes microtubule dynamics at least in part through downregulating the Cls1/Peg1/Stu1/CLASP MAP [65,66,67,68,69], and consistent with this notion, overproduction of Cls1/Peg1 is capable of rescuing the *cut7* ts mutant as *pka1∆* is [64]. Cls1/Peg1 is shown to promote microtubule bundling [70]. It is possible that enhanced bundling activity by overproduced Cls1/Peg1 would help convert the monopolar spindle in the *cut7* mutation to a bipolar spindle, leading to rescue of this mutant. 

## 4. Outward Force Generators in the Absence of Kinesin-5

Cells without Kinesin-5 become viable if Kinesin-14 is defective (e.g., *cut7∆pkl1∆*) or if microtubules are destabilised (e.g., *cut7∆* treated with microtubule-destabilising drugs). This finding poses the following important question: how do bipolar spindles assemble in the absence of Kinesin-5-mediated outward force? Detailed genetic and cell biological analyses have unravelled this puzzle; at least 11 gene products are capable of generating outward forces in place of Kinesin-5 (Table 2). 

### 4.1. Outward Forces Exerted by Kinesin-6

One possibility of the survival of *cut7∆pkl1∆* cells is that the other kinesin motors exert an outward force in place of Kinesin-5, thereby promoting spindle bipolarity. The fission yeast genome contains in total nine genes encoding kinesin motors. Genetic crosses indicate that only one kinesin, Klp9, becomes essential when combined with *cut7∆pkl1∆*; *cut7∆pkl1∆klp9∆* triple mutants are inviable. Klp9 belongs to the N-kinesin Kinesin-6. Interestingly, like Cut7 it moves on the microtubule towards the plus end and forms homotetramers, thereby crosslinking antiparallel microtubules [71,72]. Previous work showed that this kinesin accumulates at the spindle midzone upon the onset of anaphase B and promotes spindle elongation during late mitosis [71], though it appears to also play additional roles during earlier stages of mitosis [73,74,75]. Detailed analysis shows that Klp9 accelerates spindle elongation only during anaphase B in both wild type and *cut7∆pkl1∆* cells and that inviable *cut7∆pkl1∆klp9∆* cells are in fact capable of assembling bipolar spindles. However, these spindles are shorter comparted to those in wild type or *cut7∆pkl1∆* cells, and upon mitotic exit the nucleus and chromosomes are intersected by the cytokinetic actomyosin contractile ring and the septum, resulting in cell death imposed by a catastrophic “cut” (cell untimely torn) phenotype [72,76]. 

How do the two N-kinesins, Kinesin-5/Cut7 and Kinesin-6/Klp9, act in concert to drive spindle elongation in wild-type cells? Differential localisation patterns of these two kinesins on the spindle microtubule may give us a clue. While Cut7 mainly accumulates near the SPB, Klp9 is localised exclusively to the spindle midzone where antiparallel microtubules interdigitate (Figure 3A). Given these different localisations, we posit that during anaphase B, Cut7 crosslinks mainly parallel microtubules near the SPBs, while Klp9 bundles antiparallel microtubules at the spindle midzone (Figure 3B).

### 4.2. Outward Forces Exerted by the Microtubule Crosslinker and Stabiliser

Fission yeast cells are capable of forming nearly normal bipolar spindles in the presence of only Kinesin-6 Klp9, which acts in spindle elongation later in mitosis [64]. How then could spindle bipolarity be established in the first place under this condition? It transpires that two conserved MAPs, Ase1/PRC1 [77,78,79,80] and Cls1/Peg1/Stu1/CLASP [65,66,68,69], in concert, play an indispensable role in this process. Ase1 and Cls1/Peg1 bundle and stabilise antiparallel spindle microtubules (Figure 4). Notably, theoretical modelling supports bipolar spindle assembly by these two factors; Brownian dynamics–kinetic Monte Carlo simulations show that Ase1 and Cls1/Peg1 activity are sufficient for initial bipolar spindle formation [64,68,69,81,82].

Unlike in other species, *C. elegans* does not require Kinesin-5/BMK-1 for bipolar spindle formation [83]. During embryonic division of this organism, the spindle midzone could produce outward forces in concert with cortical pulling forces, thereby promoting chromosome segregation [18]. Interestingly, in this process, the midzone components including SPD-1/Ase1 and CLASP play vital roles in force generation, though SPD-1 seems to antagonise CLASP-mediated spindle elongation [18,84]. Taken together, the spindle midzone could produce outward forces in which the microtubule crosslinking and stabilising MAPs are key players. 

### 4.3. Outward Forces Exerted by Microtubule Polymerases

Alp14 and Dis1 belong to a conserved MAP family of XMAP215/Stu2/TOG that catalyses microtubule polymerisation [51,52,53,54,55,56,85,86]. As aforementioned, Alp7 forms a stable complex with Alp14 and targets the Alp14 microtubule polymerase to the SPB upon mitotic onset [49,57]. Genetic analysis indicates that any of triple deletion mutants, *cut7∆pkl1∆alp14∆*, *cut7∆pkl1∆dis1∆* or *cut7∆pkl1∆alp7∆*, are inviable. SPB-localising Csi1 and Csi2 are required for Alp7 localisation to the mitotic SPBs [87,88,89]. Consistent with this, deletion of either *csi1* or *csi2* is lethal in combination with *cut7∆pkl1∆*. Temperature sensitive *cut7∆pkl1∆alp7* cells display monopolar spindles or very short spindles (<0.5 μm) that fail to elongate [90]. These results suggest that in *cut7∆pkl1∆* cells outward forces are generated through microtubule polymerisation, in which the growing plus ends of the microtubule push the SPB, leading to separation of the SPBs (Figure 5A). 

Mutations in *dis1*, *alp7* or *alp14* rescue the *cut7* ts mutation, while rather contradictorily, the same mutations become indispensable in the *cut7∆pkl1∆* background; defects in the microtubule polymerisation confer both positive and negative impacts on the *cut7* mutation. In a single *cut7* mutation, microtubule polymerisation is negative for cell survival, while in *cut7∆pkl1∆*, it plays a positive role. This illuminates a remarkable mechanistic plasticity of bipolar spindle assembly; cells could generate either inward or outward forces using microtubule polymerases in a context-dependent manner. 

Intriguingly, in interphase fission yeast cells the nucleus is centred by pushing forces that are generated as growing cytoplasmic microtubules hit the cell tip at each end [91,92] (Figure 5B). Hence, pushing forces generated through the polymerising microtubule plus ends play important roles in both interphase and mitosis; nuclear positioning during interphase and SPB separation/bipolar spindle assembly during mitosis. The generation of an outward pushing force by the polymerising microtubule plus end is widely observed in other systems. For instance, during embryonic divisions of animal cells the plus ends of astral microtubules physically interact with and push the cell cortex, and this force ensures proper spindle positioning [95,96]. 

### 4.4. Outward Forces Exerted by the Kinetochore and Sister Chromatid Cohesion

Recent work has identified a fourth class of outward force generators [97]. This force is elicited through the kinetochore, a several-MDa–sized proteinaceous structure assembled on a specialised region of the chromosome called the centromere [98]. The spindle microtubule attaches to the kinetochore for accurate sister chromatid segregation. Mutations in genes encoding the conserved kinetochore component (Nuf2) [99] or those required for centromere-mediated sister chromatid cohesion (Rad21/Scc1/Mcd1 and Swi6/HP1) [100,101,102,103] confer a severe synthetic growth defect to *cut7∆pkl1∆*. These triple mutant cells impair proper spindle assembly and display largely monopolar spindles. These results show that the kinetochore is captured by the plus end of the spindle microtubule, thereby producing outward forces that support bipolar spindle assembly (Figure 6) [97]. 

It is noteworthy that a similar result was reported in human cells, in which stable kinetochore–microtubule attachment plays a crucial role in centrosome separation [104] and becomes essential for maintenance of the bipolar spindle in the absence of Kinesin-5 [105]. Taking all these findings together, the balance between inward and outward forces generated by opposing motor proteins, multiple MAPs, the kinetochore and sister chromatid cohesion underlies mechanisms by which bipolar spindles are formed and maintained. 

### 4.5. Outward Force Generation by Kinesin-12 in Human Cells

In fission yeast, Kinesin-6/Klp9 collaborates with Kinesin-5/Cut7 to generate an outward force. However, Klp9 acts in spindle elongation only during late mitosis irrespective of the presence or absence of Cut7, and furthermore, Klp9 cannot be substituted for Cut7 [64,90]. By contrast, in human cells a similar role appears to be executed by the N-kinesin Kinesin-12/KIF15/HKLP2, which does not exist in yeasts. It has been shown that Kinesin-12 functions redundantly with Kinesin-5 to promote spindle bipolarity [59,106,107,108], and curiously the overproduction of Kinesin-12 can drive bipolar spindle assembly even when Kinesin-5 activity is fully inhibited. This indicates that Kinesin-12 has the potential to execute all essential functions of Kinesin-5. Therefore, functions of fission yeast Kinesin-6/Klp9 and human Kinesin-12/KIF15/HKLP2 appear similar but are mechanistically different. 

## 5. Force Generation in Human Prophase Cells

Human cells undergo centrosome separation through two temporally distinct pathways, the prophase pathway and the prometaphase pathway [109,110,111]. By contrast, yeasts have adopted only the prometaphase pathway. Recent analysis has uncovered key players acting in the prophase pathway and their individual roles [110]. Duplicated centrosomes in human cells remain closely linked during interphase, in a process called centrosome cohesion. Centrosome cohesion is maintained through dual mechanisms. The first mechanism depends upon a structural linker composed of two proteins, Rootletin and C-NAP1. This linker physically joins the two centrosomes in a side-by-side configuration. The second mechanism involves the Kinesin-14/KIFC3-mediated inward force. KIFC3 forms homotetramers and interconnects a special centrosome-associated microtubule network, thereby producing pulling forces towards the centrosomes. Upon mitotic entry, KIFC3 is inactivated by the NEK2 protein kinase, which also promotes the dissolution of the linker [110]. Antagonising outward forces are produced by Kineisn-5/KIF11 in both prophase and prometaphase pathways. 

We contemplate that the differences between human beings and yeasts, in which human cells have developed more complex regulatory mechanisms, stem from different modes of mitosis; an open mitosis in higher eukaryotes vs. a closed mitosis in yeasts. As the nuclear membrane disassembles upon mitotic onset, human cells have acquired an additional regulatory process (the prophase pathway), which ensures the temporal order of bipolar spindle assembly to be synchronised with mitotic onset. Implementation of dual, redundant pathways might be also beneficial for the robustness of the system, disrupting one pathway would not result in a catastrophic impact on spindle formation and therefore the fidelity of chromosome segregation. 

## 6. Force Generation in the Acentrosomal Cells

In higher eukaryotes and plants, the bipolar spindle is formed independent of the centrosome through the pathway referred to as the acentrosomal pathway [112,113,114]. Historically, the acentrosomal pathway is extensively characterised in vitro using extracts prepared from *Xenopus* oocytes, where the Ran GTPase acts as a master regulator [115,116]. This pathway is functional in vivo in several cell types which are naturally devoid of centrosomes (e.g., vertebrate oocytes and plants) or animal somatic cells which are experimentally (chemically, genetically or physically) manipulated to eliminate their centrosomes [117,118,119]. Interestingly, in this acentrosomal pathway, both in vitro and in vivo, Kinesin-5 also plays a major role in the formation of antiparallel microtubule bundles and the generation of an outward force. However, in contrast with the centrosome-dependent pathway, Kinesin-14 and Dynein appear to act collaboratively, rather than antagonistically, with Kinesin-5. It is reported that in the *Xenopus* oocyte and human acentrosomal cells, Kinesin-5 crosslinks the antiparallel microtubules while Kinesin-14 and/or Dynein are required for spindle pole focusing [118,120], two processes needed to establish spindle bipolarity. Therefore, the importance of the force balance for bipolar spindle formation and its underlying mechanism have not been addressed explicitly in the acentrosomal cells and await further investigation. 

## 7. Towards Cancer Therapeutics

Kinesin-5 is required for successive cell division. Furthermore, in actively growing cancer cell lines, its activity is tightly, and sometimes causally, linked to tumour progression and malignancy [121]. Accordingly, this kinesin has been deemed to be an attractive target of cancer chemotherapeutics. Indeed, several Kinesin-5 inhibitors were developed and their clinical trials conducted [122]. However, drug-resistant cell lines often emerged which hampered the clinical usage of these inhibitors [123,124,125]. To tackle this conundrum, a comprehensive understanding of in vivo Kinesin-5 functions and regulations, in addition to structural information on the interaction between Kinesin-5 and specific inhibitors [122,126], are necessary. As Kinesin-12 is essential for the survival of HeLa cells that become resistant to Kinein-5 inhibitors, the development of specific Kinesin-12 inhibitors would be important [106,108,125]. Given that destabilisation and/or the reduced dynamics of the microtubule rescues the lethality derived from Kinesin-5 inactivation [25,59], the combined treatment of Kinesin-5 inhibitors and microtubule stabilising reagents would be worth consideration. In this context, treatment with Paclitaxel (Taxol), a microtubule-stabilising drug which on its own is widely used for chemotherapy (though the underlying mechanism of its anti-cancer activity remains to be resolved) [127,128,129] or suppressing Kinesin-13 microtubule depolymerases [60], might provide a more effective treatment for cancer therapeutics.

## Figures and Tables

**Figure 1 cells-09-01154-f001:**
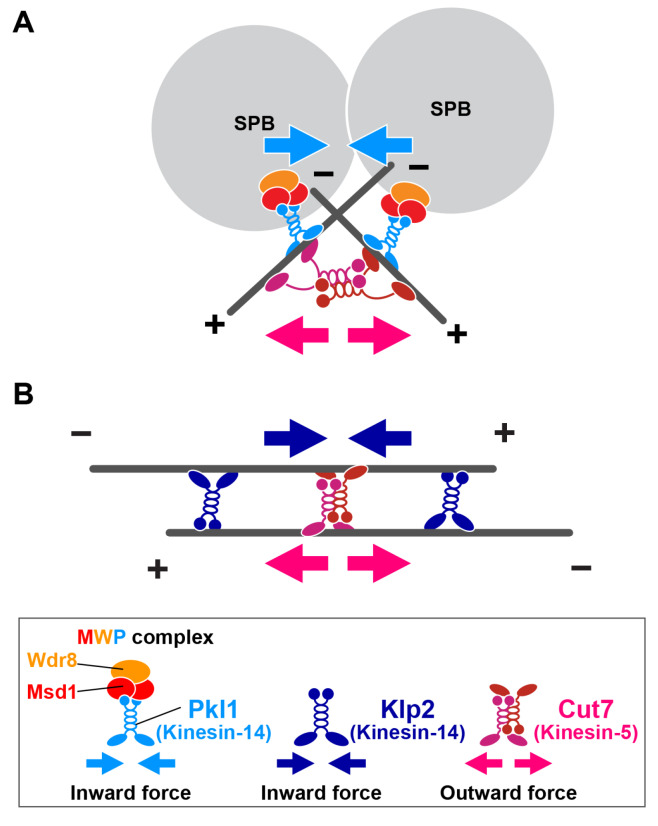
Generation of collaborative inward forces by the two Kinesin-14s, Pkl1 and Klp2. (**A**) When spindle bipolarity starts to be established upon onset of mitosis, spindle pole body (SPB)-tethered Pkl1 (the MWP complex) engages with the spindle microtubule that emanates from the opposite SPB. Minus end-directed motility of Pkl1 generates an inward force (blue arrows). This pulling force antagonises an outward force exerted by Cut7 (red arrows) that is also localised in the vicinity of the SPB and promotes interdigitation of the microtubules emanating from the two SPBs. In addition to inward force generation, the MWP complex plays a crucial role in anchoring the minus end of the spindle microtubule to the mitotic SPB as a barrier [32]. (**B**) Once bipolar spindles are formed, Klp2 and Cut7 are localised on antiparallel microtubules. These two kinesins antagonistically generate an inward force (blue arrows) and an outward force (red arrows), respectively. + and – stand for the microtubule plus and minus ends, respectively. Note that Cut7 is reportedly localised to the two other sites. One is a medial microtubule contact site. This localisation is seen when bipolar spindles are depolymerised first and then allowed to repolymerise; Cut7 promotes interpolar bundle formation [39]. The other site is the kinetochore, to which Cut7 is recruited when chromosomes are misaligned. Under this condition, Cut7 is required for chromosome gliding towards the spindle equator [40].

**Figure 2 cells-09-01154-f002:**
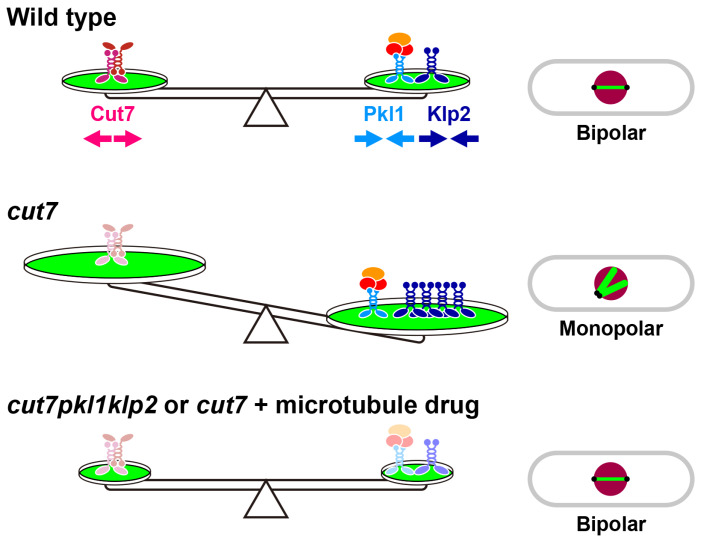
Bipolar spindle formation requires a collaborative force balance exerted by mitotic kinesins, microtubule-associated proteins (MAPs) and microtubule dynamics. In wild type cells (top), Kinesin-5/Cut7 generates an outward force (red arrows), while Kinesin-14s/Pkl1 and Klp2 generate an opposing inward force (blue arrows). Microtubule stability and dynamics promoted by a cohort of MAPs play positive roles in Klp2 activity by enhancing its localisation to the spindle microtubule, which Kinesin-5/Cut7 opposes. In *cut7* ts or *cut7∆* cells (middle), Kinesin-14-mediated inward forces dominate, leading to the formation of monopolar spindles. In double mutants between *cut7* and *pkl1*/*klp2* or the *cut7* mutant treated with microtubule-depolymerising drugs (bottom), loss of inward forces or the compromised microtubule dynamics, respectively, renders Cut7 dispensable for bipolar spindle assembly and therefore promotes survival.

**Figure 3 cells-09-01154-f003:**
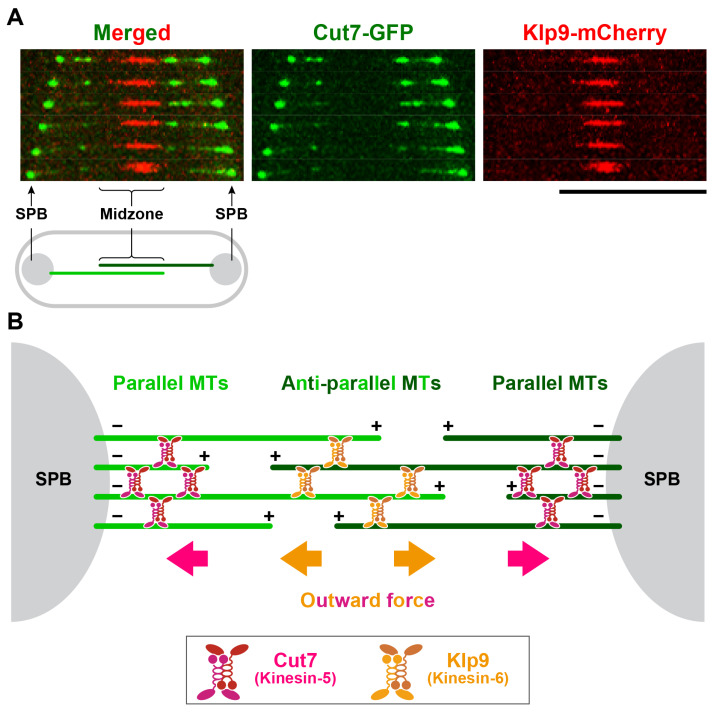
Spatially distinct localisations between Kinesin-5/Cut7 and Kinesin-6/Klp6 on anaphase B spindles. (**A**) Localisation of Cut7 and Klp9 on the spindle microtubule during anaphase B. The fission yeast strain containing *cut7-GFP* and *klp9-mCherry* (expressed from individual native promoters) were observed under fluorescence microscopy and a late mitotic cell imaged. Images were obtained using a DeltaVision microscope system (DeltaVision Elite; GE Healthcare, Chicago, IL, USA) comprising a wide-field inverted epifluorescence microscope (IX71; Olympus, Tokyo, Japan) and a Plan Apochromat 60×, NA 1.42, oil immersion objective (PLAPON 60×O; Olympus Tokyo, Japan). DeltaVision image acquisition software (softWoRx 6.5.2; GE Healthcare, Chicago, IL, USA) equipped with a charge-coupled device camera (CoolSNAP HQ2; Photometrics, Tucson, AZ, USA) was used. Time-lapse live imaging was performed after the incubation of cultures at 27 °C, in which pictures were taken at 1 min intervals as 16 sections along the z-axis at 0.2 μm intervals. The sections of images acquired at each time point were compressed into a 2D projection using the DeltaVision maximum intensity algorithm. Deconvolution was applied before the 2D projection. Captured images were processed with Photoshop CS6 (version 13.0; Adobe, San Jose, CA, USA). Scale bar, 10 μm. (**B**) A schematic showing localisations of Cut7 and Klp9 on anaphase B spindles. Cut7 bundles parallel microtubules in the vicinity of the SPB, while Klp9 bundles antiparallel microtubules at the spindle midzone. Note that Klp9 bundles antiparallel microtubules at the spindle midzone independent of its motor activity [72]. + and – stand for the microtubule plus and minus ends, respectively.

**Figure 4 cells-09-01154-f004:**
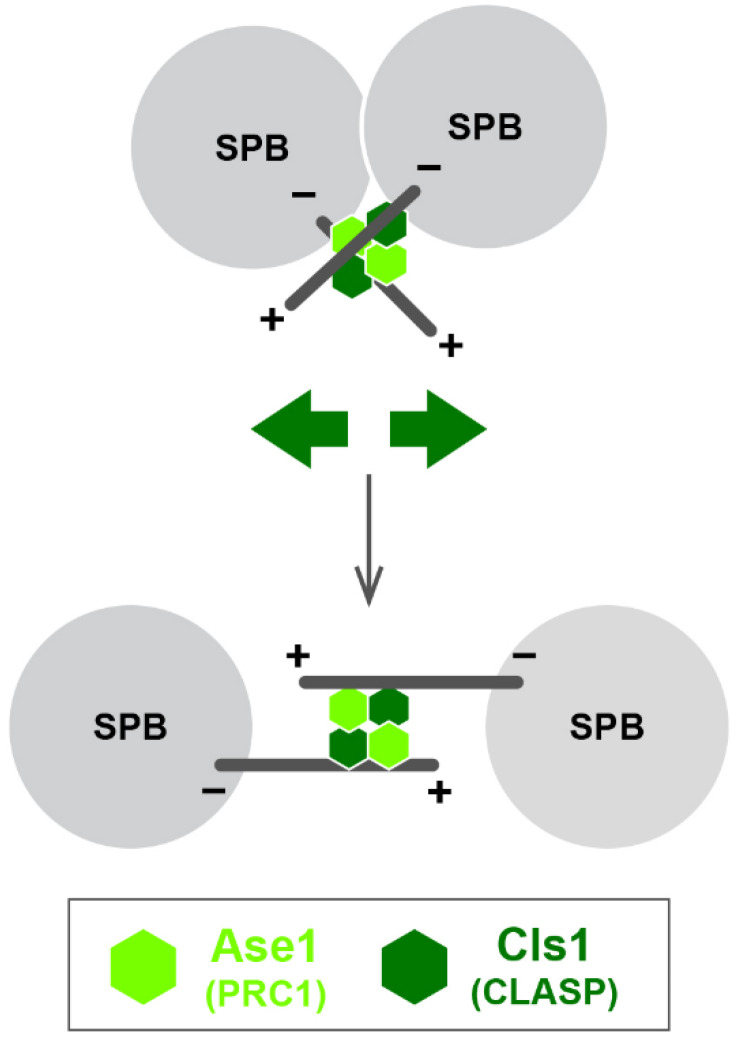
Outward force generation through the spindle midzone. Short microtubules nucleating from the two SPBs are crosslinked in an antiparallel manner and stabilised by Ase1/PRC1 and Cls1/Peg1/CLASP. This provides an outward force (green arrows) towards the SPB that is sufficient to form short bipolar spindles in the absence of Kinesin-5 and Kinesin-14 [64]. + and – stand for the microtubule plus and minus ends, respectively.

**Figure 5 cells-09-01154-f005:**
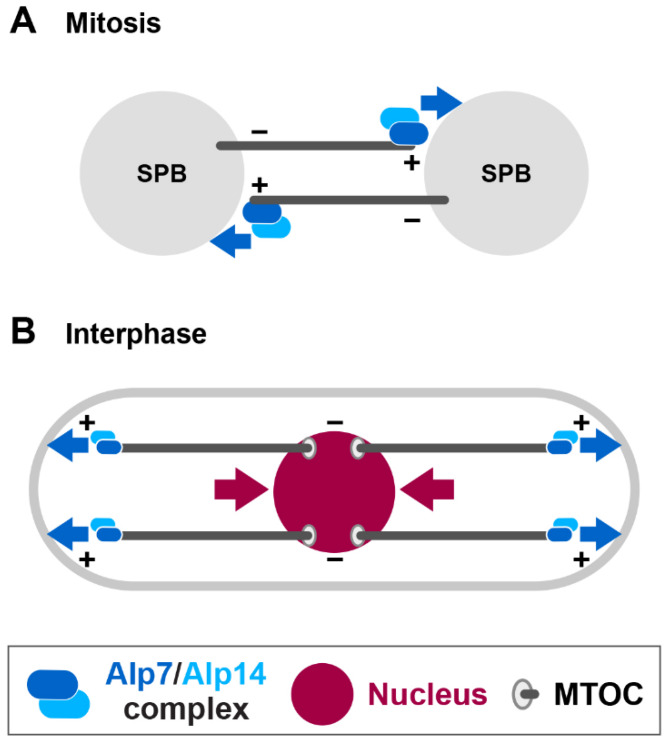
Force generation through microtubule polymerisation during interphase and mitosis. (**A**) During early mitosis, the Alp7/TACC-Alp14/TOG microtubule polymerase complex (blue ovals) is localised to the SPB (not shown) and the polymerising plus ends. Interaction of growing microtubule plus ends with the other SPB generates an outward force (blue arrows). This force is sufficient to separate the duplicated SPBs in the absence of Kinesin-5 and Kinesin-14, thereby promoting short bipolar spindle formation. (**B**) During interphase, the plus ends of the polymerising cytoplasmic microtubules reach and push the cell tip at either end (blue arrows), thereby pushing the nucleus through the opposite minus ends (deep red arrows). This allows positioning of the nucleus at the geometrical centre of the cell [91,92]. MTOC stands for microtubule organising centre, which is localised to multiple positions on the nuclear membrane during interphase [93,94]. + and – stand for the microtubule plus and minus ends, respectively.

**Figure 6 cells-09-01154-f006:**
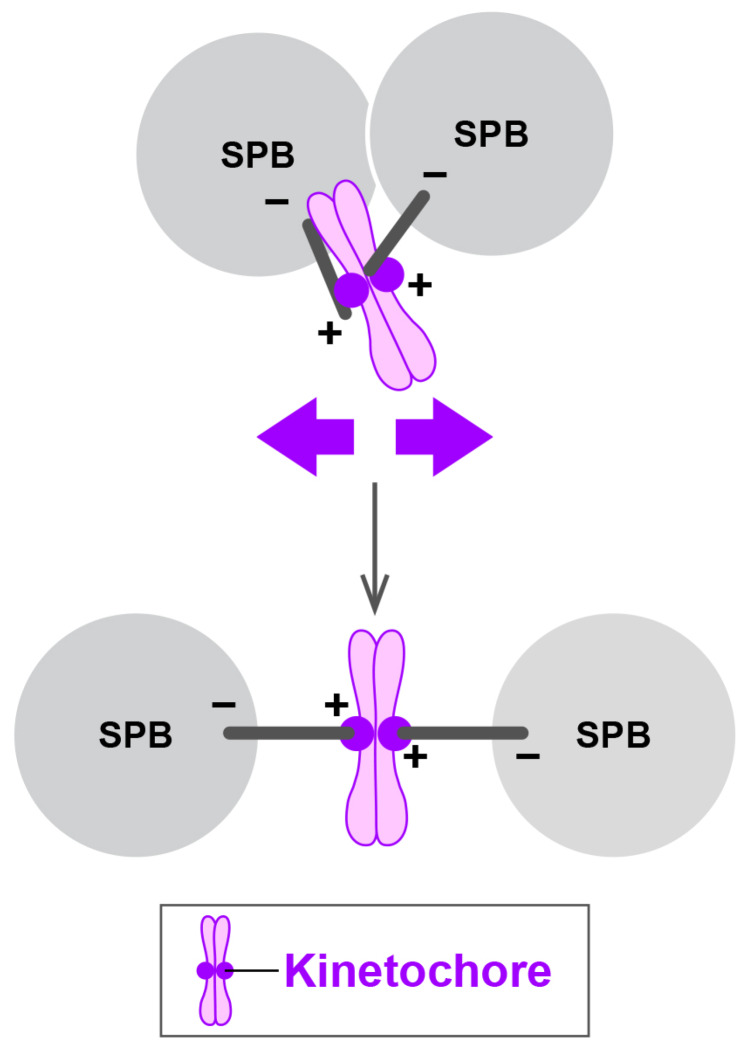
Outward force generation through the kinetochore and sister chromatid cohesion. The kinetochore and sister chromatid cohesion contribute to the generation of outward forces for SPB separation in mitosis. In the absence of Kinesin-5 and Kinesin-14, outward forces (purple arrows) derived from the kinetochore and/or sister chromatid cohesion are capable of separating the SPBs, thereby promoting bipolar spindle assembly [97]. + and – stand for the microtubule plus and minus ends, respectively.

**Table 1 cells-09-01154-t001:** List of spindle inward force generators in fission yeast.

Gene	Synonym	Protein	Homologue	Function
*pkl1* ^1^	*skf1*	Kinesin-14	HSET/KIFC1	Minus end-directed motor
*wdr8*	*skf2*	WD40 repeats	WDRB/WRAP73	A component of the MWP complex
*msd1* ^1^	*skf3*	Coiled coil	hMsd1/SSX2IP	A component of the MWP complex
*klp2*		Kinesin-14	HSET/KIFC1	Minus end-directed motor
*nda3*	*skf4*	β-tubulin	β-tubulin	Microtubule subunit
*atb2*	*skf5*	α2-tubulin	α-tubulin	Microtubule subunit
*mal3*	*skf6*	MAP	EB1	A microtubule plus-end tracking protein
*alp16*		GRIP repeats	GCP6	A component of the γ-TuC
*alp7*	*mia1*	MAP	TACC	Complex formation with Alp14
*alp14*	*mtc1*	MAP	XMAP215/Stu2/TOG	Microtubule polymerase
*dis1*		MAP	XMAP215/Stu2/TOG	Microtubule polymerase
*pka1*		Protein kinase	PKA	cAMP-dependent protein kinase

^1^ Only *pkl1* or *msd1* deletion bypasses a complete deletion of *cut7*. Mutations in the remaining genes suppress only the *cut7* temperature sensitive (ts) mutant, but not *cut7∆*. The other condition that renders *cut7∆* viable is treatment with microtubule-destabilising drugs [33].

**Table 2 cells-09-01154-t002:** List of spindle outward force generators in fission yeast.

Gene	Synonym	Protein	Homologue	Function
*klp9*		Kinesin-6	MKLP1, MKLP2	Plus end-directed motor
*ase1*		MAP	PRC1	Microtubule crosslinker
*cls1*	*peg1*	MAP	CLASP	Microtubule stabiliser/crosslinker
*alp7*	*mia1*	MAP	TACC	Complex formation with Alp14
*alp14*	*mtc1*	MAP	XMAP215/Stu2/TOG	Microtubule polymerase
*dis1*		MAP	XMAP215/Stu2/TOG	Microtubule polymerase
*csi1*		Coiled coil		Targeting Alp7 to the mitotic SPB
*csi2*		SPB localising		Targeting Csi1 to the mitotic SPB
*swi6*		Chromodomain	HP1	Heterochromatin
*rad21*		Kleisin	hRad21/Scc1/Mcd1	Cohesin
*nuf2*		Coiled coil	Nuf2	Kinetochore

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
