# Peer review of "How Essential Kinesin-5 Becomes Non-Essential in Fission Yeast: Force Balance and Microtubule Dynamics Matter"

_cells, 2020, doi:10.3390/cells9051154_

Round 1

Reviewer 1 Report

The review by Yukawa et al on the mechanism driving bipolar spindle formation in a kinesin-5 independent manner focus on work performed in fission yeast. It is a nice synthetic summary of the data obtained by this group from a screen aiming at finding suppressors for cut7 (Eg5) temperature sensitive mutants in fission yeast. They found 3 types of suppressors and review the data highlighting three mechanisms that can account for bipolar spindle formation in absence of functional kinesin 5.

It is overall an interesting review that will certainly help many researchers working in the field.

Several issues however should be addressed before publication.

1- The title does not reflect well the content of the review and is therefore a bit misleading. The review is centered on data obtained in fission yeast and this should be clear from the title. Maybe the authors could also revise the title to make more attractive?

2- It is somehow confusing that there are a few comments throughout the text on non fission yeast data but these are often incomplete or out of the scope of the review.

It may be better to keep the focus on the fission yeast data and include all the data from other systems in a new specific chapter at the end discussing similarities and differences between the yeast and human systems. This could be an interesting chapter highlighting that in some cases the proteins involved may be different but the activities are conserved. Also this would make a better context to mention the roles of dynein and kinesin depolymerases (that are not mentioned at all) (and add references). The section 4.5 on kinesin-12 could be placed there as well.

3- Chapter 6 is not well integrated in the review. It is not complete and in fact could be eliminated altogether.

4- Finally chapter 7 is not really well integrated and does not flow well after the review focused on fission yeast. It should be revised and focus more on what has been learned from the work in fission yeast to propose potentially novel therapeutic lines. It does not seem very logical to mention here the need for finding inhibitors for kinesin-12 (that does not exist in yeast) and the argument in the last part on microtubule destabilization (Lines 386-389) is not clear.

Specific comments:

1- Table 1, line 91: modify the title to ‘List of spindle inward force generators in fission yeast’

Please correct:

            - pkl1 is not a kinesin-6 but kinesin 14.

            - Klp2 is not a MAP but a kinesin 14

2- Table 2; modify the title to ‘List of spindle outward force generators in fission yeast’

3- Line 371: the sentence: “However, in contrast with the centrosome-dependent pathway, Kinesin-14 and Dynein appear to act collaboratively, rather than antagonistically, with Kinesin-5” is not supported by any reference and is quite confusing. What are the evidences to support it? In any case this whole section could be deleted (see comment above).

Reviewer 2 Report

This review is focuses on the function of kinesin-5 and how other factors contribute to mitotic spindle assembly in the absence of kinesin-5 function. A lot of the discussions are based on the results in their 2019 G3 paper, which described a suppressor screen for genes whose mutations can compensate for the loss of kinesin-5 (cut7) function. At the end, they provided a nice discussion on the implication of these results in cancer therapy using kinesin-5 as a drug target.  This review is overall very well written and I would support its publication after revision. My main criticism is that this review is too narrowly focused on fission yeast data without even mentioning some earlier results from other fungal systems such as the budding yeast and the filamentous fungus Aspergillus nidulans. My specific comments/suggestions are as follows:

  1. It would be nice to mention that kinesin-5’s mitotic function was first revealed by a genetic study in Aspergillus nidulans and cite the following studies that are closely relevant to the opposite functions of kinesin-5 and kinesin-14 and were done before the authors’ 2019 study: Saunders and Hoyt, 1992, Cell; O’Connell et al., 1993, J Cell Biol; Hoyt et al., 1993 Genetics; Wang et al., 2015, Fungal Genetics and Biol. The Hoyt et al., 1993 and the Wang et al., 2015 papers described similar suppressor screens but the mutations were only identified in genes encoding kinesin-14 (unlike the S. pombe screen that identified more loci). Could this be related to the nature of the mutant alleles used? For example, could the bimC4 mutant allele in nidulans be more like a null allele than the cut7 ts allele used in the S. pombe screen?
  2. In table 1 (Points 3-5 are also related to table 1), “kinesin-6” should be changed to “kinesin-14” (after pkl1). After klp2, “MAP” can be changed to “kinesin-14”.
  3. It is not proper to list EB1 as “Coiled-coil” under “Protein”, listing it as a “MAP” should be ok. Many proteins contain coiled-coils and EB1 also contains other domains. The same criticism also applies to Table 2 where some proteins are listed as “coiled coil”.
  4. “A plus-end tracking protein” can be changed to “A microtubule plus-end-tracking protein”. In addition, I am not sure if tubulins could be considered as MAPs- they are microtubule subunits.
  5. The authors may consider adding Stu2 beside XMAP215 because people in the budding yeast field should be more familiar with the name Stu2 (covered in Winey and Bloom, 2012, Genetics).
  6. Line 244-245, “…MAPs, Ase1/PRC1 [68, 69] and Cls1/Peg1/CLASP, in concert, play an indispensable role in this process.” It would be better to cite the original papers on budding yeast Ase1 together with S. pombe refs 68 and 69 (Pellman et al., 1995 JCB; Schuyler et al., 2003 JCB) given that Ase1 was first identified in budding yeast. A reference or references should be added after “CLASP”, and also the budding yeast name Stu1 can be added (For example, Yin et al., 2002 MBoC). Reference should be added also at the end of the sentence (Rincon et al., 2017 Nat Commu. etc)
  7. I like the discussion on “Towards cancer therapeutics”. The authors may consider adding an interesting paper that discussed the difference between using Taxol and using a kinesin-5 inhibitor for cancer therapy (Mitchison et al., 2017, Open biol.). However, this is not essential and the authors can decide on this.
